# Hyperpolarized $^{15}$N-labeled, deuterated tris (2-pyridylmethyl)amine as an MRI sensor of freely available $Zn^{2+}$

Eul Hyun Suh [1], Jae Mo Park [1,2,3], Lloyd Lumata[4], A. Dean Sherry [1,2,5] & Zoltan Kovacs [1,5 ✉]

Dynamic nuclear polarization (DNP) coupled with $^{15}$N magnetic resonance imaging (MRI) provides an opportunity to image quantitative levels of biologically important metal ions such as $Zn^{2+}$, $Mg^{2+}$ or $Ca^{2+}$ using appropriately designed $^{15}$N enriched probes. For example, a Zn-specific probe could prove particularly valuable for imaging the tissue distribution of freely available $Zn^{2+}$ ions, an important known metal ion biomarker in the pancreas, in prostate cancer, and in several neurodegenerative diseases. In the present study, we prepare the cell-permeable, $^{15}$N-enriched, $d_6$-deuterated version of the well-known $Zn^{2+}$ chelator, tris(2-pyridylmethyl)amine (TPA) and demonstrate that the polarized ligand had favorable $T_1$ and linewidth characteristics for $^{15}$N MRI. Examples of how polarized TPA can be used to quantify freely available $Zn^{2+}$ in homogenized human prostate tissue and intact cells are presented.

[1] Advanced Imaging Research Center, University of Texas Southwestern Medical Center, Dallas, TX 75390, USA. [2] Department of Radiology, University of Texas Southwestern Medical Center, Dallas, TX 75390, USA. [3] Department of Electrical and Computer Engineering, University of Texas at Dallas, Richardson, TX 75080, USA. [4] Department of Physics, University of Texas Dallas, Richardson, TX 75080, USA. [5] Department of Chemistry and Biochemistry, University of Texas Dallas, Richardson, TX 75080, USA. ✉email: zoltan.kovacs@utsouthwestern.edu

Nitrogen is one of the four most abundant elements in the human body yet there are very few spectroscopic methods available to monitor the various forms of nitrogen in biological molecules. Nitrogen-15 (0.37% natural abundance) has a favorable nuclear spin of ½ and a wide chemical shift range (900 ppm), yet [15]N NMR is much less widely used as a tool in biological systems compared to [13]C and [1]H because of its poor sensitivity[1]. However, [15]N NMR in combination with hyperpolarization techniques such as dissolution dynamic nuclear polarization (d-DNP) or parahydrogen induced polarization (PHIP) is beginning to find its way into the biological realm[2]. DNP refers to technologies that enhance the NMR signal-to-noise ratio by amplifying nuclear spin polarization via microwave driven transfer of high electron spin polarization to coupled nuclear spins at low temperatures and high magnetic fields[3]. The frozen HP sample is then rapidly dissolved with a superheated solvent to produce a solution of the hyperpolarized compound. The hyperpolarized spin state is not persistent and decays to thermodynamic equilibrium by spin-lattice (longitudinal) relaxation. Hence, unlike in conventional NMR experiments, exceedingly long $T_1$ values are advantageous for HP studies. The sensitivity improvements offered by HP makes it feasible to perform molecular/functional MR imaging of nuclei other than [1]H.

The most widely studied NMR-active nucleus for hyperpolarized metabolic imaging is [13]C. HP-[13]C-labeled tracers not only allow one to detect biological products from a starting HP substrate but also provide the opportunity to perform dynamic measurements to determine flux through specific enzyme catalyzed reactions in vivo by [13]C MRS/MRI[4–7]. The [15]N nucleus has several favorable properties for HP studies including narrow linewidths, relatively long $T_1$ values, and a much larger chemical shift range compared to [13]C. HP-[15]N-labeled compounds are particularly well suited for creating newer types of sensors that can potentially offer molecular information with high spectral resolution, low background interference, and minimal invasiveness. Successful application of hyperpolarized [15]N probes in biological systems requires [15]N-labeled compounds that have long spin-lattice ($T_1$) relaxation time and large chemical shift dispersion in response to changes in physiological parameters such as pH, redox, and concentration of reactive oxygen species or biologically relevant metal ions[8,9].

Diamagnetic $Zn^{2+}$ is an important target for imaging because these ions participate in various biochemical processes such as enzyme catalysis, neurotransmission, intracellular signaling, and antibiotic activity. It has been shown that cellular $Zn^{2+}$ plays an important role in the progression of diabetes and prostate cancer as well as in the development of various neurodegenerative disorders[10,11]. In the pancreas, $Zn^{2+}$ is essential for proper storage of insulin β-cell granules[12]. In the brain, $Zn^{2+}$ is released along with glutamate from presynaptic vesicles via calcium-dependent exocytosis. Healthy prostate tissue stores and secretes large quantities of $Zn^{2+}$ ions while malignant cells have reduced levels of $Zn^{2+}$[13–18]. For this reason alone, quantitative imaging of $Zn^{2+}$ in vivo could be a very useful diagnostic tool for the detection of malignant tissues. Although there are many cell-based optical methods for monitoring intracellular and extracellular $Zn^{2+}$, nondestructive detection of $Zn^{2+}$ levels in vivo remains a major challenge. Optical methods have potential for quantitative in vivo detection of $Zn^{2+}$ but the limited penetration depth due to attenuation and scattering of light in tissues is a major obstacle[19]. Two recent MRI studies have demonstrated that a gadolinium-based Zn-responsive contrast agent can be used to monitor $Zn^{2+}$ secretion from pancreatic islets[20,21] and prostate in mice[22]. In both tissues, secretion of $Zn^{2+}$ from intracellular stores into the extracellular space was initiated after a bolus injection of glucose. Given that the $Zn^{2+}$ content of prostate tissue is known to be much lower in prostate cancer[22], this technology has the potential to differentiate between healthy versus malignant tissue in a simple MRI exam in combination with a $Zn^{2+}$-sensitive contrast agent. However, it should be pointed out that this test does not detect total $Zn^{2+}$ content of prostate tissue but rather only the $Zn^{2+}$ ions released in response to glucose. Hence, an imaging method that provided a direct measure of total $Zn^{2+}$ content could perhaps be even more informative. The detection limit of a typical Gd-based MRI contrast agent is also limited to around 50 μM so this could become a limiting factor in some tissues[11].

There have been previous reports of HP-based sensors for detection of metal ions including [13]C-EDTA[23] and [129]Xe-based $Zn^{2+}$ sensors[24]. However, these do have limitations for in vivo use such as significant line broadening of the ligand [13]C signals after binding to $Zn^{2+}$[23] and the lack of sufficient chemical shift differences between the $Zn^{2+}$, $Mg^{2+}$, and $Ca^{2+}$ complexes[24]. A recent study demonstrated that HP [1-[13]C]cysteine can detect $Zn^{2+}$ in biological samples at physiological concentrations[25]. In that system, the [13]C resonance of [1-[13]C]cysteine shifted downfield and broadened upon addition of $Zn^{2+}$ so quantitative detection of $Zn^{2+}$ could be problematic because the signal position is not only dependent upon freely available $Zn^{2+}$ levels but also the sensor concentration as well. It was also not clear in these previous studies whether these reporter systems would report total tissue $Zn^{2+}$ or specifically extracellular $Zn^{2+}$.

Here we report a HP-[15]N based probe, tris(2-pyridylmethyl) amine (TPA) (Fig. 1) for the detection and quantification of total freely available $Zn^{2+}$ in biological samples. TPA is a tripodal ligand that has excellent selectivity for $Zn^{2+}$ over other common biological cations. Most optical and Gd-based $Zn^{2+}$ responsive agents contain a sensing moiety structurally derived from TPA[22,26,27]. This compound is known to distribute across cell membranes[27] so in principle should detect freely available tissue $Zn^{2+}$ in all compartments and not just $Zn^{2+}$ ions released from cells in response to a biological stimulus[22].

## Results

**[15]N-enriched TPA design.** To evaluate TPA as a prototype $Zn^{2+}$ imaging sensor, we first performed NMR experiments on HP TPA containing natural abundance level of [15]N. There are two types of nitrogen atoms in TPA, the pyridine N-atoms and the central aliphatic tertiary N-atom. [15]N NMR spectra showed that there is a substantial chemical shift difference in both the pyridine (50 ppm upfield) and the central [15]N (20 ppm upfield) signals after addition of $Zn^{2+}$ ions (Supplementary Fig. 1). However, the pyridine [15]N signals relaxed more rapidly (the $T_1$ was not measured) than the central [15]N signal so, based on these observations, the tertiary nitrogen was chosen as the optimal site for [15]N enrichment.

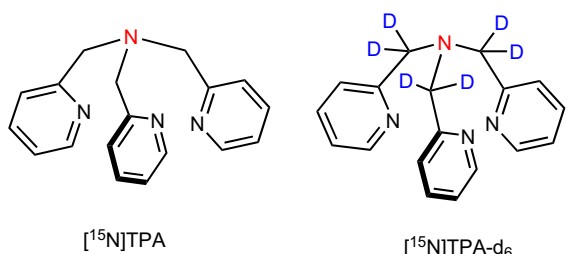

[15N]TPA                    [15N]TPA-d₆

**Fig. 1 Structure of [15]N-labeled tris(2-pyridylmethyl)amine derivatives discussed in this work.** Tripodal ligands based on tris(2-pyridylmethyl) amine form $Zn^{2+}$ complexes with high stability and excellent selectivity.

**Fig. 2 Synthesis of $^{15}$N-labeled TPA derivatives.** The key intermediate in the synthesis is [$^{15}$N]phthalimde (**1**), conveniently prepared by heating a mixture of phthalic anhydride and [$^{15}$N]ammonium acetate.

The $^{15}$N-labeled version of TPA was synthesized starting from $^{15}$N-labeled ammonium acetate and 2-chloromethyl pyridine or 1-(chloromethyl-$d_2$) pyridine as outlined in Fig. 2 with an overall yield of 25% and 61%, respectively. As nearby protons can provide an efficient relaxation mechanism for $^{15}$N by dipole–dipole interaction, we also prepared a $^{15}$N-TPA derivative in which deuterium was substituted for each proton in the proximity of the $^{15}$N nucleus (Fig. 2). This modification is expected to significantly increase the $^{15}$N $T_1$ relaxation time.

**Dynamic nuclear polarization of the $^{15}$N probes**. The compounds polarized well in a HyperSense polarizer using standard DNP conditions with OX063 trityl radical as polarizing agent to a level of $8 \pm 2\%$ ($23,795 \pm 6950$) and $6 \pm 2\%$ ($18,164 \pm 6237$) polarization for [$^{15}$N]TPA and [$^{15}$N]TPA-$d_6$, respectively, after dissolution in PBS buffer (Supplementary Fig. 2). Both ligands showed an identical 20 ppm upfield shift in their $^{15}$N resonances upon addition of Zn$^{2+}$ (Fig. 3a and Supplementary Fig. 2). The $T_1$ of the non-deuterated [$^{15}$N]-TPA and Zn$^{2+}$-[$^{15}$N]TPA species were $25.9 \pm 1.6$ s and $17.9 \pm 0.1$ s[28,29], respectively (Fig. 3b). However, as expected, deuteration of the methylene groups resulted in a substantially prolonged $T_1$ values for [$^{15}$N]TPA-$d_6$ and Zn$^{2+}$-[$^{15}$N]TPA-$d_6$: $70.9 \pm 1.1$ ($p < 0.05$ compared to $T_1$ values of [$^{15}$N]TPA) and $57.0 \pm 2.3$ ($p < 0.05$ compared to $T_1$ values of Zn$^{2+}$-[$^{15}$N]TPA) at 9.4 T, respectively. The detection limits of these polarized samples were evaluated by varying the [Zn$^{2+}$] in samples of HP-[$^{15}$N]TPA-$d_6$ and observing the $^{15}$N signals in buffered solutions. These experiments showed that the detection threshold of Zn$^{2+}$ using HP-[$^{15}$N]TPA-$d_6$ was ~5 μM (Fig. 3c).

**MRSI of HP $^{15}$N-TPA-$d_6$.** In a proof of principle MR imaging experiment, $^{15}$N CSI of phantoms containing HP-[$^{15}$N]TPA-$d_6$ in the absence and presence of Zn$^{2+}$ was collected to evaluated the feasibility of spectroscopic imaging of Zn$^{2+}$ (Fig. 4). HP-[$^{15}$N]TPA-$d_6$, polarized for 2 h, was injected into three NMR tubes

containing different concentrations of [Zn$^{2+}$] (Fig. 4a). Axial $^{15}$N chemical shift images of these tubes were acquired using a standard CSI sequence at 9.4 T. The $^{15}$N resonances of the free and Zn-bound [$^{15}$N]TPA-$d_6$ were observed at 40 ppm and 20 ppm reflecting [$^{15}$N]TPA-$d_6$ (Fig. 4b, c) and Zn$^{2+}$-[$^{15}$N]TPA-$d_6$ (Fig. 4b, d), respectively. The $^{15}$N data were reconstructed and analyzed using MATLAB (Mathworks, Natick MA, USA).

**HP-[$^{15}$N]TPA-$d_6$ to measure freely available Zn$^{2+}$ in biological sample**. We next tested the feasibility of using $^{15}$N-NMR and [$^{15}$N]TPA-$d_6$ to quantify Zn$^{2+}$ levels in tissue and intact cell samples. First, known amounts of HP-[$^{15}$N]TPA-$d_6$ (1 mM) were added to HEPES buffered samples containing physiologically relevant concentration of Zn$^{2+}$ ranging from 0 to 200 μM and $^{15}$N spectra were recorded using a single 45 degree pulse. Given that the total concentration of [$^{15}$N]TPA-$d_6$ was known, the area of the Zn$^{2+}$-[$^{15}$N]TPA-$d_6$ signal relative to the total $^{15}$N signal provides a direct readout of total Zn$^{2+}$ (Supplementary Eq. 1). Figure 5a shows there is an excellent correlation between [Zn$^{2+}$] as measured by HP-$^{15}$N NMR versus the analytically determined Zn$^{2+}$ levels ($R^2 = 0.99$). To illustrate the feasibility of using [$^{15}$N]TPA-$d_6$ as a sensor of free tissue Zn$^{2+}$, an HP sample of [$^{15}$N]TPA-$d_6$ was mixed with a fresh surgically resected human prostate tissue homogenate. The tissue was homogenized as described in the Methods section to facilitate rapid mixing prior to the addition of freshly prepared HP-[$^{15}$N]TPA-$d_6$ (final concentration of 4.2 mM). Sequential $^{15}$N NMR spectra were then recorded every 2 s to monitor decay of the HP signals (Supplementary Fig. 4). Figure 5b shows the first $^{15}$N NMR spectrum collected within seconds after mixing HP-[$^{15}$N]TPA-$d_6$ with the tissue sample demonstrating that the ligand is in excess with respect to the total amount of Zn$^{2+}$ present in the sample. Quantification of Zn$^{2+}$ as described in the Supplementary Methods indicated that the total concentration of freely available Zn$^{2+}$ in this particular tissue sample was 66 μM. This value is in good agreement with the Zn concentration (71 μM) measured by ICP-MS. It should be

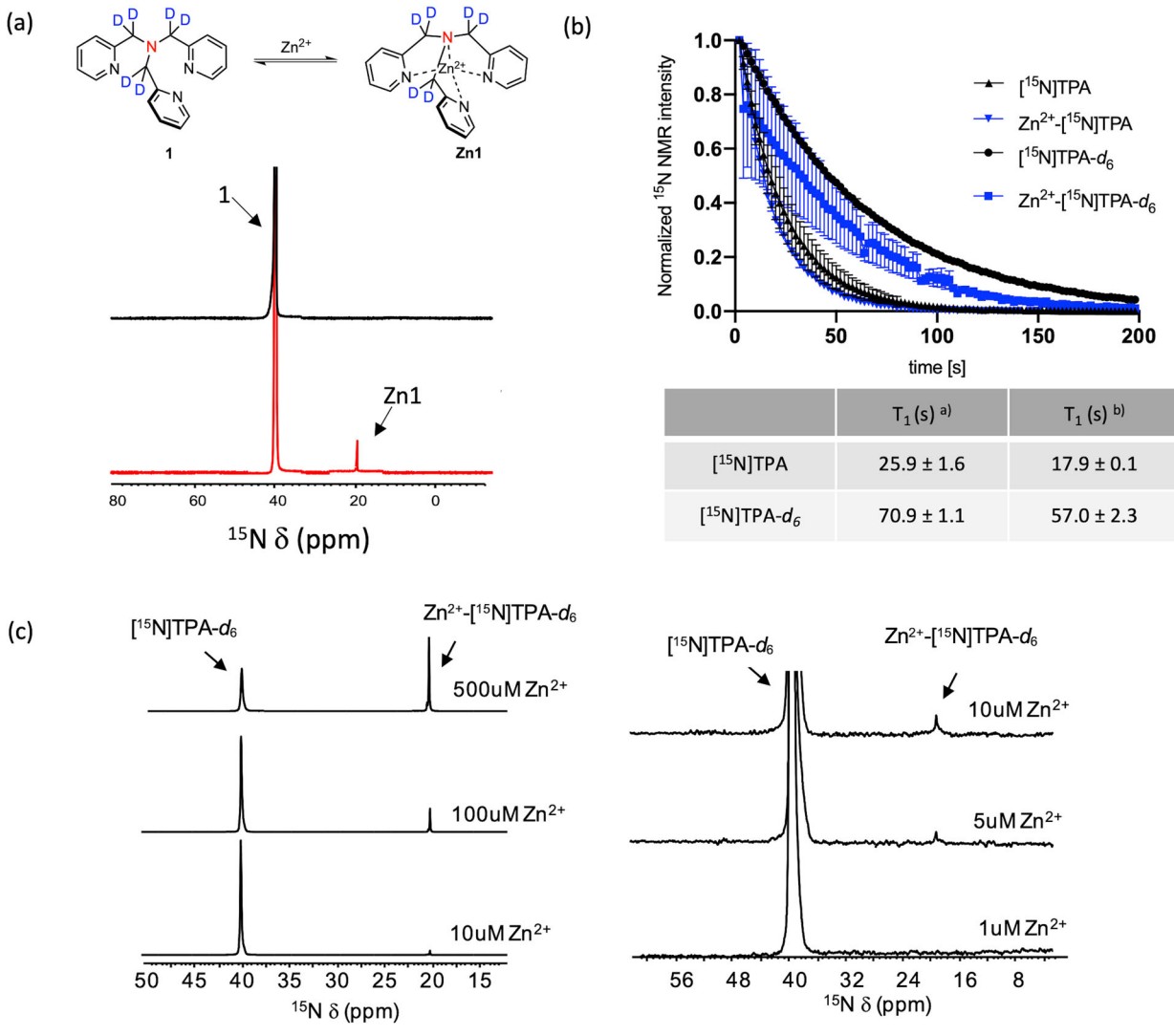

**Fig. 3 DNP-¹⁵N MR experiments of [¹⁵N]TPA derivatives. a** ¹⁵N NMR chemical shift of [¹⁵N]TPA-$d_6$ in the absence and presence of $Zn^{2+}$ (0.25 eq).
**b** Sequential [¹⁵N]TPA and [¹⁵N]TPA-$d_6$ spectrum decay and $T_1$ relaxation of free ligand [a)] and $Zn^{2+}$ complexes [b)] measured at 9.4 T, 298 K, pH 6.8.
**c** Single scan of ¹⁵N NMR spectra of HP [¹⁵N]TPA-$d_6$ (1.2 mM) with various concentration of $Zn^{2+}$.

noted, however, that homogenization destroys all cellular compartments and thus, this experiment does not demonstrate that HP-[¹⁵N]TPA-$d_6$ can be used to measure intracellular $Zn^{2+}$. To examine whether HP-[¹⁵N]TPA-$d_6$ can detect the free $Zn^{2+}$ in intact cell, we performed HP experiments in which HP-[¹⁵N] TPA-$d_6$ was added to $Zn^{2+}$ spiked human prostate epithelial (PNT1A) cells after the extracellular zinc was washed away[30]. Note that spiking is necessary to achieve Zn levels in cultured cells that mimic in vivo situation. Figure 5c, d shows the time-dependent ¹⁵N spectra and the first spectrum, respectively, after HP-[¹⁵N]TPA-$d_6$ was mixed with intact PNT1A cells. Based on the area of the $Zn^{2+}$-[¹⁵N]TPA-$d_6$ signal relative to the total ¹⁵N signal and the known total concentration of [¹⁵N]TPA-$d_6$ (2.8 mM), the $Zn^{2+}$ concentration in these samples were calculated to be 118 and 123 μM ($n = 2$) (Supplementary Methods). These values agree well with the result of ICP-MS analysis (151 and 146 μM, respectively).

## Discussion
In this study, we demonstrate that ¹⁵N-labeled TPA can be used as hyperpolarized magnetic resonance $Zn^{2+}$ probe to detect the freely available $Zn^{2+}$. In general, hyperpolarized chemical probes

are designed as a combination of sensing and signaling moieties[8]. In this case, the signaling part is the HP-¹⁵N-enriched nucleus and the sensing part consists of a tetradentate tripodal ligand structure that ensures high selectivity for the analyte of interest, $Zn^{2+}$. We have chosen tris(2-pyridylmethyl) amine (TPA) for Zn-sensing moiety because it forms stable complexes with $Zn^{2+}$ and displays excellent selectivity over most endogenous metal ions. A deuterated version of [¹⁵N]TPA was also synthesized to prolong the life-time of the hyperpolarized ¹⁵N reporter atom. While the chemistry involved in the synthesis is fairly straightforward, special attention was given in the synthesis design to finding suitable starting materials for the introduction of the ¹⁵N and ²H label and selecting optimal reaction conditions that give reasonable yields in each step to maximize the overall yield of the final, ¹⁵N-enriched and deuterated product. ¹⁵N-labeled phthalimide, conveniently synthesized by melting together ¹⁵N ammonium acetate and phthalimide, satisfied these requirements to afford the final product [¹⁵N]TPA and [¹⁵N]TPA-$d_6$ in reasonable overall yield.

It is well-known that all four nitrogen donor atoms of TPA coordinate to $Zn^{2+}$ and this results in an upfield shift of the ¹⁵N chemical resonance[26]. The fact that separate signals were observed for the free ligand and the $Zn^{2+}$ complex throughout a

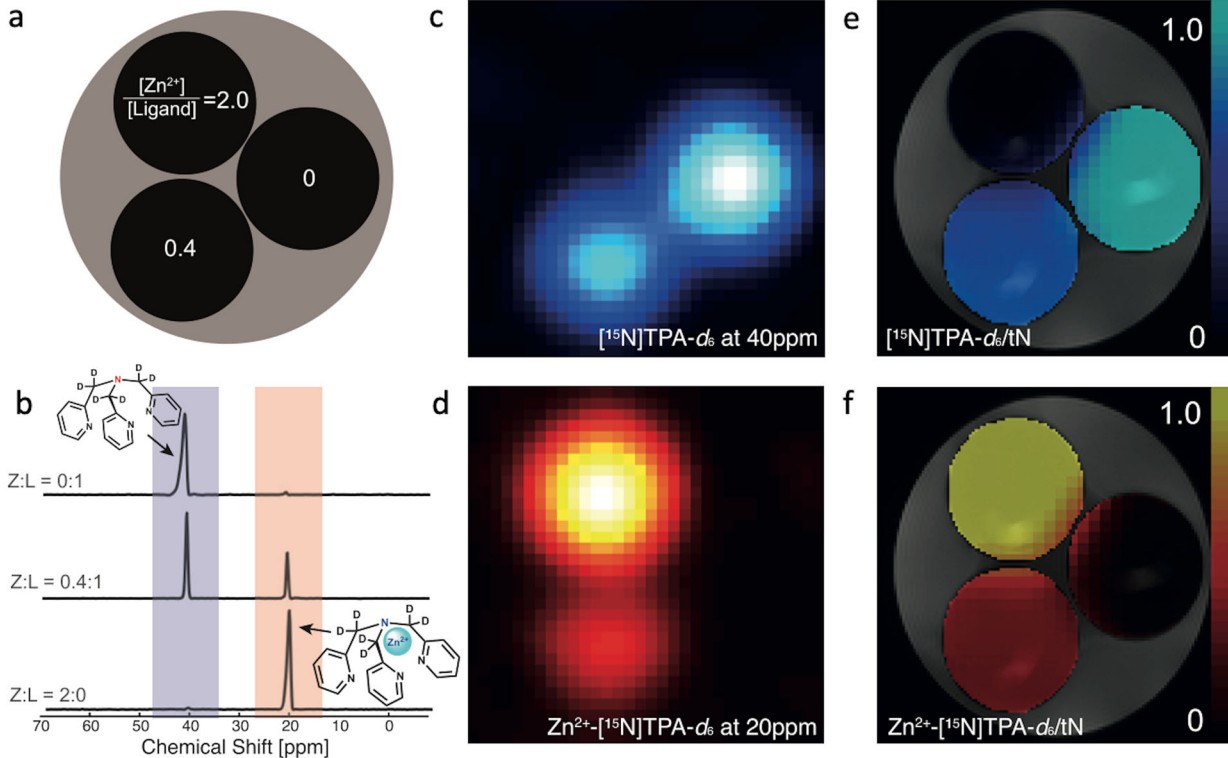

**Fig. 4 $^{15}$N CSI of HP [$^{15}$N]TPA-$d_6$ in HEPES buffered phantom solutions containing different ratios of Zn$^{2+}$/[$^{15}$N]TPA-$d_6$. a** A diagram of the phantom arrangement; **b** $^{15}$N spectra at different Zn$^{2+}$ to [$^{15}$N]TPA-$d_6$ ratios; **c** $^{15}$N images of free, uncomplexed [$^{15}$N]TPA-$d_6$ (40 ppm); **d** images of the Zn$^{2+}$-[$^{15}$N]TPA-$d_6$ complex (20 ppm). **e** and **f** show the ratiometric images of each species, normalized by the total $^{15}$N signal (tN).

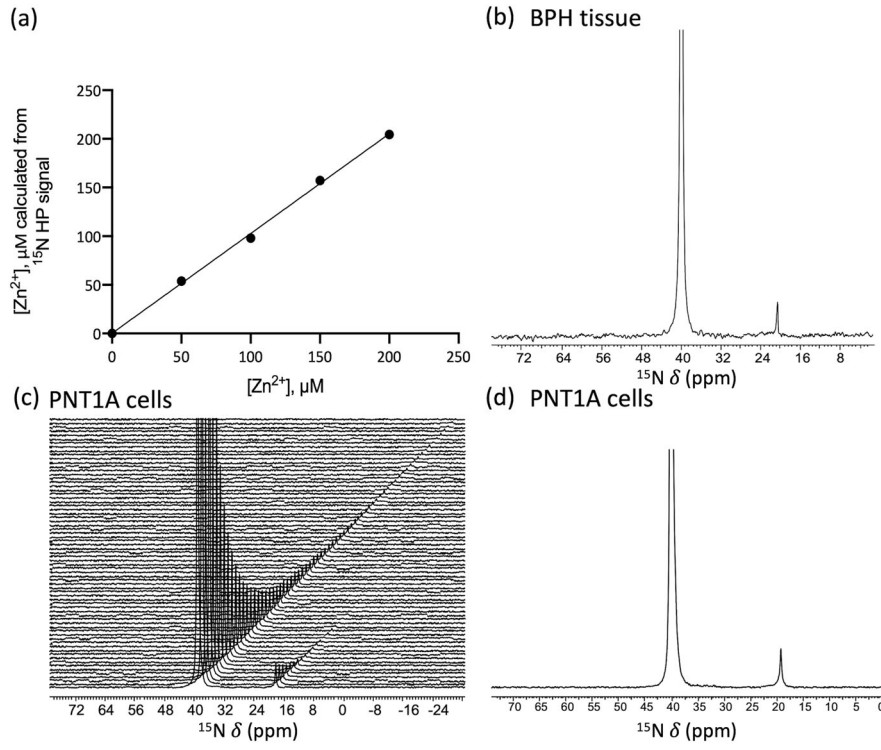

**Fig. 5 Quantitative detection of [Zn$^{2+}$] in biological samples. a** A plot of [Zn$^{2+}$] as determined by HP versus the analytically known [Zn$^{2+}$] in each sample. **b–d** quantitative detection of [Zn$^{2+}$] in a surgical specimen of human benign prostatic hyperplasia (BPH) tissue and human prostate epithelial PNT1A cells. Spectra of free ligand (40 ppm) and Zn$^{2+}$ complexes (20 ppm) were recorded every 2 s using 30° and 45° flip angle for BPH tissue and PNT1A cells. **b** the first $^{15}$N spectrum collected after dissolution and mixing on a 500 mg sample of BPH tissue after adding 4.2 mM HP-[$^{15}$N]TPA-$d_6$. (9.4 T, 298 K and pH 7.0). **c** Time-dependent $^{15}$N spectra collected on a sample of PNT1A cells (~65 × 10$^6$ cells) after adding 2.8 mM HP TPA-$d_6$ and **d** the first $^{15}$N spectrum. (9.4 T, 298 K and pH 7.4).

TPA/$Zn^{2+}$ titration shows that exchange between the two species is slow on the NMR timescale (Fig. 3a). This is an advantage over HP-[1-$^{13}$C]cysteine for detection of $Zn^{2+}$ because the ratio of the bound versus unbound resonance areas provide a direct readout of total $Zn^{2+}$ (Fig. 3a). Like [1-$^{13}$C]cysteine, [$^{15}$N]TPA is also selective for $Zn^{2+}$, but unlike cysteine, its $pK_a$ values are below 7, which eliminates interference from protonation of the free unbound ligand. In addition, while $^{15}$N has lower NMR sensitivity compared to $^{13}$C because of its lower gyromagnetic ratio, $^{15}$N typically displays much longer $T_1$ values and sharper NMR resonances than $^{13}$C. Moreover, deuterated version of [$^{15}$N]TPA further elongated the $^{15}$N $T_1$. In general, the dominant mechanism for spin-lattice relaxation of $^{15}$N includes both dipolar and chemical shift anisotropy (CSA) contributions. The major source of dipolar relaxation is the dipole–dipole interaction of $^{15}$N with $^1$H spins present in the molecule. The dipolar relaxation rate ($R_d$) is independent of the $B_0$ field strength ($R_d \sim \frac{\gamma_N^2 X \gamma_{1H}^2}{r^6}$, where $\gamma$ is gyromagnetic ratio and $r$ is the distance between the $^{15}$N nucleus and the dipolarly coupled $^1$H), while the CSA relaxation rate ($R_{CSA}$) increases with $B_0$ ($R_{CSA} \sim \gamma_N^2 \Delta\sigma^2 B_0^2$, where $B_0$ is the magnetic field and $\Delta\sigma$ is the magnetic shielding anisotropy)[31–36]. The $R_d$ can be minimized by substituting $^2$H (deuterium), which has much lower $\gamma$ than $^1$H, for all nearby protons[33,37] The contribution of $R_{CSA}$ can be reduced by increasing the molecular symmetry and/or performing the HP experiments at low field[38]. The significant increase in $T_1$ upon deuteration indicated that the dominant relaxation mechanism for $^{15}$N-TPA was dipolar relaxation. Measurements of the $^{15}$N $T_1$ at other field strengths (1, 3, and 9.4 T) yielded similar $T_1$ values confirming that the chemical shift anisotropy contribution was indeed negligible as expected for a such a symmetrical molecule (TPA and related tripodal ligands have a $C_3$ symmetry). (Supplementary Table 1)[39].

Chemical shift imaging (CSI), widely used in HP-$^{13}$C studies, provides both spectral and spatial information[40]. Here, $^{15}$N CSI experiments were performed on phantoms in a vertical bore high-resolution spectroscopy magnet equipped with gradients. [$^{15}$N]TPA-$d_6$ and its $Zn^{2+}$-complex were easily distinguished with good spatial resolution and sensitivity due to the large, 20 ppm $^{15}$N NMR chemical shift separation between the free ligand and the Zn complex. This result clearly demonstrates that HP-[$^{15}$N]TPA-$d_6$ has the potential to map the distribution of freely available $Zn^{2+}$ (defined as all $Zn^{2+}$ except that tightly bound to metalloproteins). We also demonstrated that the $Zn^{2+}$ concentration in phantoms can be assessed using the integrals of the free and Zn-bound [$^{15}$N]TPA-$d_6$ signals. In principle, protonation of TPA would alter the effective binding constant for $Zn^{2+}$ (the conditional stability of the complex) but, fortunately, the $pK_a$ values of TPA are 2.55, 4.35, 6.17[41], so the ligand is largely deprotonated at pH 7.4 and little proton competition exists. This was also evident in $^{15}$N NMR spectra of TPA, whose chemical shift did not change between pH 6.4 and 8.0. The TPA binding unit is commonly used in optical probe designs because of its high affinity and specificity for $Zn^{2+}$ (log $K_{ZnTPA} = -11.0$)[26,27,42].

HP studies confirmed that there is no interference from $Ca^{2+}$ ions (2 mM) at biologically relevant pH values (Supplementary Fig. 3a). In addition, a 1:1 binding stoichiometry was confirmed by thermal $^{15}$N NMR spectroscopy (Supplementary Fig. 3b) and a detection limit of 1~5 $\mu$M $Zn^{2+}$ (Fig. 3c) is well-below the intracellular concentration of $Zn^{2+}$ in secretory cells ($\mu$M to mM)[14,43,44]. The feasibility of using the [$^{15}$N]TPA-$d_6$ as a $Zn^{2+}$ sensor in biological samples was demonstrated using freshly isolated human benign prostate hyperplasia (BPH) homogenates prepared from biopsy samples as well as human prostate epithelial (PNT1A) cells. ICP-MS measurements of the Zn concentration verified that

[$^{15}$N]TPA-$d_6$ could accurately report $Zn^{2+}$ levels in these biological samples. The successful $Zn^{2+}$ detection in intact PNT1A cells also demonstrated that the cellular uptake of HP-[$^{15}$N]TPA-$d_6$, a known membrane permeable Zn chelator[27], fast compared to HP-[$^{15}$N] signal decay.

In general, hyperpolarized $^{15}$N for in vivo imaging has remained largely unexplored at this point primarily due to the technical challenges associated with $^{15}$N. First, conventional preclinical and clinical MR scanners are typically not equipped for imaging low gamma nuclei. Hyperpolarization of exogenous $^{15}$N-labeled substrates overcomes the MR sensitivity problem, but other scanner-associated requirements in hardware (RF coils and frequency synthesizer) and software remain for $^{15}$N imaging. Moreover, the low Larmor frequency requires stronger gradient coil performance to achieve sharp localization (e.g., slice selection) and high-resolution image acquisition.

In conclusion, we have developed hyperpolarized $^{15}$N-labeled TPA derivatives for use as an imaging sensor of freely available $Zn^{2+}$ in a biological sample. The tertiary $^{15}$N atom in deuterated TPA was found to have favorable properties including a long $T_1$ value and a sharp $^{15}$N resonance that displays a large chemical shift difference upon complexation of $Zn^{2+}$. It was shown that HP-[$^{15}$N]TPA-$d_6$ can detect $Zn^{2+}$ in the low $\mu$M range with no interference from protons or other endogenous metal ions. The probe was successfully used to quantify free Zn levels in human prostate tissue homogenate and intact human prostate epithelial cells. Since total $Zn^{2+}$ in the prostate is known to drop dramatically in malignant tissue, the current results demonstrate that the use of HP-[$^{15}$N]TPA-$d_6$ to measure freely available $Zn^{2+}$ in prostate tissues in vivo during progression of prostate cancer would likely be diagnostically informative. The cytotoxicity of TPA due to its strong binding to $Zn^{2+}$, which can lead to zinc depletion, could limit its application for in vivo studies[27]. This, however, could be ameliorated by the design of second generation agents with weaker Zn-binding constant.

## Methods

**General remarks**. All reagents and solvents were purchased from commercial sources and used as received without further purification unless stated otherwise. Zinc analyses (ICP-MS) were performed by Galbraith Laboratories, Inc. $^1$H, $^{13}$C, and $^{15}$N NMR spectra were recorded using a 9.4 T Varian/Agilent VNMRS400 NMR spectrometer operating at 400, 100, and 40 MHz, respectively. LCMS experiments were carried out on a Waters Alliance LC system with a Atlantis T3 column, (C18, 5 $\mu$, 250 × 4.6 mm), connected to a Waters diode array UV detector, and an electrospray ionization mass spectrometer (ESI MS) using a Waters Qtof-MS-XEVO ESI positive mode for detection.

**General procedures for dynamic nuclear polarization (DNP) NMR studies**. DNP was performed using a HyperSense commercial polarizer (3.35 T, Oxford Instruments Molecular Biotools, UK). The TPA ligands were dissolved in glycerol-water matrix (50:50 w/w) containing 2 mM Gd chelate (ProHance®) and 15 mM trityl radical polarizing agent (OX063, GE Healthcare, UK). The final concentrations of the substrates in the glassing matrix was ~0.5 M. Sample volumes for DNP varied from 18 $\mu$L to 50 $\mu$L. The polarization was performed at 1.05 K with 94.112 GHz microwave irradiation for 2 h and the frozen polarized samples were subsequently rapidly dissolved in a phosphate buffered solution (10 mM phosphate, pH~7) or HEPES buffered solution in a 10-mm NMR tube. $^{15}$N NMR spectra were acquired with a 9.4 T Varian vertical bore microimager (Varian, USA) using flip angle of 5° to 45° depending on the experiment with repetition time (TR) of 2 s. All $^{15}$N chemical shifts were externally referenced to [$^{15}$N]ammonium chloride (0 ppm). The $^{15}$N spectra were processed using ACD/SpecManager (ACD Labs, Canada). The binding stoichiometry experiment was performed using thermal $^{15}$N NMR spectrometer (45° flip angle with repetition time (TR) of 250 s. For chemical structure characterization, $^1$H, $^{13}$C, and $^{15}$N NMR spectra were recorded on Varian VNMRS direct drive console spectrometer operating at 400, 100, and 40 MHz, respectively. The $^{15}$N spin-lattice relaxation times ($T_1$) were measured by following the decay of the HP-$^{15}$N magnetization over time by applying a small flip angle pulse. The $T_1$ values were calculated using the following equation.

$$I(t) = I_0 \sin\theta(\cos\theta)^{t/TR} exp^{-t/T_1}$$

Where, $I(t)$ is the signal intensity at time $t$, $I_0$ is the initial magnetization, TR is

repetition time, and $\theta$ is the flip angle of radiofrequency (rf) pulse used to monitor the hyperpolarization signal.

## Synthesis of $^{15}$N-labeled tris(2-pyridylmethyl)amine, [$^{15}$N]TPA.

**[$^{15}$N]Phthalimide (1):** [$^{15}$N]Ammonium acetate (6.5 mmol) was mixed with phthalic anhydride (5.4 mmol) and stirred for 4 h at 160 °C. After the reaction mixture cooled to room temperature, the white solid was washed with cold water, filtered, and dried in vacuum to afford [$^{15}$N]phthalimide as white powder (0.73 g, 91% yield). $^1$H NMR (DMSO-$d_6$, 400 MHz) δ (ppm) = 11.27 (1H, br s, NH), 7.81 (4H, s, phthalic); $^{13}$C NMR (DMSO-$d_6$, 100 MHz) δ (ppm) = 169.21 (d, $^1J_{CN}$ = 13 Hz, C=O), 134.28, 132.56 (d, $^2J_{CN}$ = 7 Hz, CC=O), 122.91; $^{15}$N NMR (DMSO-$d_6$, 40 MHz) δ = 153.51; LCMS(ESI): m/z calc. for [M + H]$^+$ = 149.0369, found [M + H]$^+$ = 149.0351.

**[$^{15}$N](Pyridine-2-ylmethyl)phthalimide (2)[45,46]:** [$^{15}$N]Phthalimide (645 mg, 4.3 mmol) was dissolved in dimethylformamide (10 mL) and potassium carbonate (1.5 g, 10.9 mmol) and 2-(chloromethyl)pyridine hydrochloride (749 mg, 4.6 mmol) were added. The reaction mixture was stirred for 12 h at 50 °C. After removal of the solvent by rotary evaporation, water was added to precipitate the product and to remove inorganic salts. The crude product was filtered and dried in vacuum to afford [$^{15}$N](pyridine-2-ylmethyl)phthalimide as a white solid (1.03 g, 98% yield). $^1$H NMR (DMSO-$d_6$, 400 MHz) δ (ppm) = 8.43 (1H, d, J = 4 Hz, pyridyl), 7.90 (2H, s, phthalic), 7.87 (2H, s, phthalic), 7.76 (1H, t, J = 8 Hz, pyridyl), 7.40 (1H, d, J = 8 Hz, pyridyl), 7.26 (1H, t, J = 4 Hz, pyridyl), 4.91 (2H, s, NCH$_2$); $^{13}$C NMR (DMSO-$d_6$, 100 MHz) δ = 166.78 (d, $^1J_{CN}$ = 13 Hz, C=O), 155.11, 149.07, 136.88, 134.56, 131.69 (d, $^2J_{CN}$ = 8 Hz, CC=O), 123.23, 122.52, 121.28, 42.11 (d, $^2J_{CN}$ = 10 Hz, CH$_2$$^{15}$N-); $^{15}$N NMR (DMSO-$d_6$, 40 MHz) δ = 154.45; LCMS(ESI): m/z calc. for [M + H]$^+$ = 240.0791, found [M + H]$^+$ = 240.0600.

**[$^{15}$N]-2-Aminomethyl pyridine (3):** [$^{15}$N](Pyridine-2-ylmethyl)phthalimide (850 mg, 3.6 mmol) was dissolved in 10% hydrochloric acid (30 mL) and the solution was stirred for 12 h at 80 °C. The hydrochloric acid was removed by rotary evaporation and the residue was dissolved in 20% sodium hydroxide (5 mL) and extracted with dichloromethane (50 mL). The organic layer was dried over anhydrous sodium sulfate, filtered, and concentrated by rotary evaporation. The residue was dried in vacuum to afford [$^{15}$N]-2-aminomethyl pyridine as a yellowish oil (349 mg, 89% yield). $^1$H NMR (CDCl$_3$, 400 MHz) δ (ppm) = 8.52 (1H, d, J = 4 Hz, pyridyl), 7.61 (1H, t, J = 8 Hz, pyridyl), 7.24 (1H, d, J = 8 Hz, pyridyl), 7.12 (1H, t, J = 4 Hz, pyridyl), 3.95 (2H, s, NCH$_2$); $^{13}$C NMR (CDCl$_3$, 100 MHz) δ (ppm) = 161.67, 149.32, 136.62, 121.90, 121.31, 47.70 (d, $^1J_{CN}$ = 3 Hz, C-$^{15}$NH$_2$); $^{15}$N NMR (CDCl$_3$, 40 MHz) δ (ppm, referenced to [$^{15}$N]NH$_4$Cl) = 98.90; LCMS (ESI): m/z calc. for [M + H]$^+$ = 110.0736, found [M + H]$^+$= 110.0750.

**[$^{15}$N]Tris(2-pyridylmethyl)amine (4):** [$^{15}$N]-2-Aminomethyl pyridine (348 mg, 3.2 mmol) was dissolved in acetonitrile (15 mL) and potassium carbonate (1.8 g, 12.8 mmol) and 2-(chloromethyl)pyridine hydrochloride (1.1 g, 6.4 mmol) were added. The reaction mixture was stirred for 12 h at ambient temperature. After filtration, the crude product was dissolved in dichloromethane and filtered to remove potassium carbonate. The organic layer was dried over anhydrous sodium sulfate, filtered, and dried in vacuum. The oily residue was purified by column chromatography on silica gel with dichloromethane and methanol (97:3) as eluent followed by recrystallization from ethyl ether/hexane to afford [$^{15}$N]-tris(2-pyridylmethyl)amine as light yellowish solid (291 mg, 31% yield). $^1$H NMR (CDCl$_3$, 400 MHz) δ (ppm) = 8.51 (3H, d, J = 4 Hz, pyridyl), 7.61 (3H, t, J = 8 Hz, pyridyl), 7.55 (3H, d, J = 8 Hz, pyridyl), 7.11 (3H, t, J = 4 Hz, pyridyl), 3.87 (6H, s, NCH$_2$); $^{13}$C NMR (CDCl$_3$, 100 MHz) δ (ppm) = 159.38, 149.17, 136.51, 123.05, 122.08, 60.22 (d, $^1J_{CN}$ = 5 Hz, –CH$_2$-$^{15}$N-); $^{15}$N NMR (CDCl$_3$, 40 MHz) δ (ppm, referenced to [$^{15}$N]NH$_4$Cl) = 39.71; LCMS(ESI): m/z calc. for [M + 1]$^+$ = 292.1580, found [M + H]$^+$= 292.1456.

## Synthesis of $^{15}$N-labeled tris(2-pyridylmethyl-$d_2$)amine, [$^{15}$N]TPA-$d_6$.

**2-Pyridinemethan-$d_2$-ol (5):** Sodium borodeuteride (4.2 g, 99.2 mmol) was added slowly to a deuterated methanol solution (18 mL) of ethyl picolinate (5.0 g, 33.1 mmol) at 0 °C for 40 min. The reaction mixture was stirred for 12 h at ambient temperature. After removal of the solvent by rotary evaporation, the residue was stirred with saturated potassium carbonate solution (100 mL) for 1 h and then extracted with chloroform. The organic layer was dried over anhydrous sodium sulfate, filtered, and dried in vacuum to afford the product as white solid (3.22 g, 88% yield). $^1$H NMR (CDCl$_3$, 400 MHz) δ (ppm) = 8.26 (1H, d, J = 4 Hz, pyridyl), 7.47 (1H, t, J = 8 Hz, pyridyl), 7.24 (1H, d, J = 8 Hz, pyridyl), 6.96 (1H, t, J = 8 Hz, pyridyl), 5.72 (s, br, –OH); $^{13}$C NMR (CDCl$_3$, 100 MHz) δ = 160.20, 148.05, 136.71, 121.93, 120.61, 63.53 (p, $J_{CD}$ = 21 Hz, CD$_2$); $^2$H NMR (CDCl$_3$, 61 MHz) δ = 4.57 (2D, CD$_2$OH). LCMS(ESI): m/z calc. for [M + H]$^+$ = 112.0731, found = 112.0596.

**2-(Chloromethyl-$d_2$)pyridine (6):** 2-(Chloromethyl-$d_2$)pyridine was prepared as previously described[47]. Briefly, 2-pyridinemethan-$d_2$-ol (3.2 g, 28.0 mmol) was added dropwise to a solution of thionyl chloride (16.7 g, 140.2 mmol) in dichloromethane (20 mL) in an ice bath. The reaction mixture was refluxed for 3 h and then stirred at room temperature for 12 h. The product was precipitated by adding ethyl ether (20 mL). The precipitation was filtered and dried in vacuum to afford 2-(chloromethyl-$d_2$)pyridine hydrochloride salt as pale yellow powder (3.56 g, 77% yield as the HCl salt). $^1$H NMR (CDCl$_3$, 400 MHz) δ (ppm) = 8.66 (1H, d, J = 4 Hz, pyridyl), 8.44 (1H, t, J = 8 Hz, pyridyl), 8.02 (1H, d, J = 8 Hz,

pyridyl), 7.89 (1H, t, J = 8 Hz, pyridyl); $^{13}$C NMR (CDCl$_3$, 100 MHz) δ (ppm) = 150.71, 146.68, 140.85, 127.36, 126.61, 38.77 (p, $J_{CD}$ = 24 Hz, CD$_2$); $^2$H NMR (CDCl$_3$, 61 MHz) δ (ppm) = 5.00 (2D, CD$_2$Cl). LCMS(ESI): m/z calc. for [M + H]$^+$ = 130.0393, found = 130.0232.

**[$^{15}$N](2-Pyridinelmethyl-$d_2$)phthalimide (7):** It was synthesized from 2-(chloromethyl-$d_2$)pyridine hydrochloride as described for $^{15}$N-(pyridine-2-ylmethyl)phthalimide. [$^{15}$N]phthalimide (1.5 g, 10.1 mmol) was dissolved in dimethylformamide (18 mL) and potassium carbonate (3.5 g, 25.3 mmol) and 2-(chloromethyl-$d_2$) pyridine (1.8 g, 11.1 mmol) was added and the reaction mixture was stirred for 12 h at 50 °C. After removal of the solvent, water was added to the crude product to remove the inorganic salts. The precipitated product was filtered and dried in vacuum to afford [$^{15}$N](2-pyridinemethyl-$d_2$)phthalimide as white powder (2.39 g, 98% yield). $^1$H NMR (CDCl$_3$, 400 MHz) δ (ppm) = 8.51 (1H, d, J = 4 Hz, pyridyl), 7.86 (1H, s, phthalic), 7.85 (1H, s, phthalic), 7.72 (1H, s, phthalic), 7.71 (1H, s, phthalic), 7.61 (1H, t, J = 8 Hz, pyridyl), 7.26 (1H, d, J = 8 Hz, pyridyl), 7.14 (1H, t, J = 4 Hz, pyridyl); $^{13}$C NMR (CDCl$_3$, 100 MHz) δ (ppm) = 168.20 (d, $^1J_{CN}$ = 13 Hz, C=O), 155.35, 149.75, 136.74, 134.12, 132.27, 123.56, 122.58, 121.67, 42.64 (m, J = 25.5 Hz, -$^{15}$NCD$_2$); $^2$H NMR (CDCl$_3$, 61 MHz) δ (ppm) = 4.95 (2D, -$^{15}$NCD$_2$); $^{15}$N NMR (CDCl$_3$, 40 MHz) δ (ppm, referenced to [$^{15}$N]NH$_4$Cl) = 154.14; LCMS(ESI): m/z calc. for [M + H]$^+$ = 242.0916, found = 242.0682.

**[$^{15}$N]-2-(Aminomethyl-$d_2$)pyridine (8):** This intermediate was synthesized from [$^{15}$N](2-pyridinemethyl-$d_2$)phthalimide as described for $^{15}$N-2-aminomethyl pyridine. A solution of [$^{15}$N](2-pyridinemethyl-$d_2$)phthalimide (2.2 g, 9.3 mmol) in 10% hydrochloride (45 mL) was stirred for 12 h at 80 °C. After removal of the hydrochloric acid, the residue was dissolved in 20% sodium hydroxide (13 mL) and extracted with dichloromethane. The organic layer was dried with anhydrous sodium sulfate, filtered, and dried in vacuum to afford [$^{15}$N]-2-(aminomethyl-$d_2$) pyridine as a pale yellow oil (877 mg, 85% yield). $^1$H NMR (CDCl$_3$, 400 MHz) δ = 8.20 (1H, d, J = 4 Hz, pyridyl), 7.26 (1H, t, J = 8 Hz, pyridyl), 6.91 (1H,d, J = 8 Hz, pyridyl), 6.77 (1H, t, J = 8 Hz, pyridyl); $^{13}$C NMR (CDCl$_3$, 100 MHz) δ (ppm) = 161.45, 148.60, 135.84, 121.11, 120.53, 47.70 (m, J = 20 Hz, -$^{15}$NH$_2$CD$_2$); $^2$H NMR (CDCl$_3$, 61 MHz) δ (ppm) = 3.56 (2D, -$^{15}$NH$_2$CD$_2$); $^{15}$N NMR (CDCl$_3$, 40 MHz) δ (ppm, referenced to [$^{15}$N]NH$_4$Cl) = 106.98; LCMS(ESI): m/z calc. for [M + H]$^+$ = 112.0862, found = 112.0730.

**[$^{15}$N]Tris(2-pyridylmethyl-$d_2$)amine (9):** [$^{15}$N]-2-(Aminomethyl-$d_2$)pyridine (535 mg, 4.9 mmol) was dissolved in acetonitrile (15 mL) and potassium carbonate (2.7 g, 19.6 mmol) and 2-(chloromethyl-$d_2$)pyridine (1.6 g, 9.8 mmol) were added. The reaction mixture was stirred for 12 h at 50 °C. The inorganic salts were removed by filtration and organic layer was evaporated by rotary evaporation. The residue was dissolved in hot ethyl acetate and filtered to remove insoluble impurities. The solution was concentrated by rotary evaporation and the oily residue was purified by recrystallization in ethyl ether to afford $^{15}$N-tris(2-pyridylmethyl-$d_2$)amine as pale yellow solid (920 mg, 81% yield). $^1$H NMR (CDCl$_3$, 400 MHz) δ = 8.35 (3H, d, J = 4 Hz, pyridyl), 7.47 (3H, t, J = 8 Hz, pyridyl), 7.42 (3H, d, J = 4 Hz, pyridyl), 6.95 (3H,t, J = 4 Hz, pyridyl); $^{13}$C NMR (CDCl$_3$, 100 MHz) δ = 159.05, 148.60, 136.12, 122.73, 121.74, 59.02 (pd, $J_{CD}$ = 21 Hz, -$^{15}$NH$_2$CD$_2$); $^2$H NMR (CDCl$_3$, 61 MHz) δ = 3.73 (2D, -$^{15}$NCD$_2$); $^{15}$N NMR (CDCl$_3$, 40 MHz) δ (ppm) = 40.47; LCMS(ESI): m/z calc. for [M + H]$^+$ = 298.1957, found = 298.1693.

## Hyperpolarized $^{15}$N MRS studies with human benign prostatic hyperplasia (BPH) tissue samples.

Human PBH tissue (approximately 500 mg) was homogenized in 1 mL 10% aqueous solution of NP-40 (nonyl phenoxypolyethoxylethanol) using a tissue homogenizer (PowerGen 500, Fisher Scientific) in an ice bath. The resulting mixture was further lysed using Ultrasonic Homogenizer (3 × 10 s). The pH was adjusted to pH~3 and allowed to stand for 10 min in an ice bath. It was then centrifuged for 10 min at 12,000 rpm. The supernatant was collected and the pH was adjusted to around 6 for the HP experiment. The homogenized prostate tissue (1 mL) was placed in 10 mm NMR tube. Hyperpolarized [$^{15}$N]TPA-$d_6$ was injected into the tube within 10 s after dissolution with PBS (final concentration of 4.2 mM, pH 7.0). HP $^{15}$N MRS studies were performed 9.4 T using 30° flip angle with repetition time (TR) of 2 s.

## Hyperpolarized $^{15}$N MRS studies with human prostate epithelial cells (PNT1A-WT).

Normal prostate epithelial cells (PNT1A) were cultured in a 150 mm culture dish with RPMI-1640 medium (Sigma, R8758) supplemented with 10% of fetal bovine serum (Sigma, F2442) and 20 units of penicillin-streptomycin (Sigma, P0781). When the cells reached ~90% confluence, they were washed twice with PBS (5 mL) and supplemented with zinc. Zn-supplementation was performed with 15 μM zinc pyrithione (Sigma, H6377) in assay buffer (114 mM NaCl, 4.7 mM KCl, 1.2 mM KH$_2$PO$_4$, 2.5 mM CaCl$_2$, 1.16 mM MgSO$_4$, 3 mM glucose and 20 mM HEPES, pH 7.4). After 10 min incubation with zinc pyrithione at 37 °C, the buffer was removed and the cells were washed three times with 5 mL PBS. Assay buffer (4 mL) was added to cell culture plate and the cells (~65 × 10$^6$) were collected using a cell scraper (Sigma, CLS3008). After centrifuging at 1000 rpm for 5 min, the supernatant was removed and the cell pellets were resuspended in assay buffer to achieve 1 m L of final volume. The cells were suspended in 1 mL of assay buffer and placed in 10 mm NMR tube. Hyperpolarized [$^{15}$N]TPA-$d_6$ was injected into the tube within 10 sec following the dissolution with assay buffer (final concentration

of 2.8 mM, pH 7.4). HP $^{15}$N MRS studies in cells were performed 9.4 T using 45° flip angle with repetition time (TR) of 2 s.

**$^{15}$N chemical shift imaging (CSI)**. Phantom imaging with [$^{15}$N]TPA-$d_6$ was performed on a 9.4 T Agilent (Varian) vertical bore microimager (Varian, USA). The phantom consisted of three 8-mm NMR tubes each containing different concentrations of zinc chloride inserted into a 22-mm NMR tube containing deionized water (10 mL). The axial imaging plane was positioned near the center of the phantom. The acquisition of $^{15}$N CSI started 10 s after the HP sample transfer was completed. CSI parameter: CSI2 sequence (Agilent VnmrJ 4 Imaging, USA), field of view (FOV) = 40 × 40 mm; TR = 200 ms; TE = 1.30 ms; flip angle = 30°; NA = 1, matrix = 16 × 16. The $^{15}$N CSI imaging data were reconstructed and analyzed using MATLAB (Mathworks, Natick MA, USA).

**Statistical analysis**. Statistical significance was evaluated by an unpaired $t$-test ($\alpha = 0.05$, two-tailed, homoscedastic) using GraphPad Prism version 8.0 (Graph-Pad Software, Inc., La Jolla CA, USA). All data are presented as mean ± SD.

**Associated content**. The Supplementary Information is available for general experimental considerations, detailed synthetic procedures, characterization of [$^{15}$N]TPA derivatives, and additional supporting data (PDF).

## Data availability
The authors declare that all data supporting the findings of this study are available within the paper and Supplementary Information.

## Code availability
The MATLAB scripts used for image reconstruction in the study will be available from the author upon request.

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

## Acknowledgements

We thank Dr. Douglas Strand and Dr. Jer-Tsong Hsieh at the Department of Urology, the University of Texas Southwestern Medical Center, Dallas, TX for providing the human prostate tissue samples and PNT1A-WT cell. We acknowledge partial financial support for this work from the National Institutes of Health (R37-HR034557 and P41-EB015908), the Texas Institute for Brain Injury and Repair, the Robert A. Welch Foundation (AT-584, AT-1877-03242018 and I-2009-20190330), and Collaborative Biomedical Research Award (CoBRA).

## Author contributions

Z.K. conceived the idea. E.H.S., A.D.S., and Z.K. designed the research; E.H.S. and L.L. performed the preliminary HP-stubiodies. E.H.S. and J.M.P. performed the hyperpolarized $^{15}N$ imaging experiment; E.H.S. and Z.K. performed the synthesis TPA derivatives and performed hyperpolarized $^{15}N$ spectroscopy in tissue; E.H.S., A.D.S., and Z.K. performed the HP studies in biological samples. E.H.S. A.D.S., and Z.K. analyzed the $^{15}N$ NMR spectra data; E.H.S., J.M.P., A.D.S., and Z.K. wrote the manuscript.

## Competing interests

The authors declare no competing interests.
