## [Peer Review File · Communications Chemistry]

Reviewers' comments:

Reviewer #1 (Remarks to the Author):

This manuscript by Kovacs and coworkers reports the detection of free available Zn²⁺ in tissues with a hyperpolarized ¹⁵N-labeled MRI sensor. A new and characteristic ¹⁵N NMR signal at precise chemical shift was produced upon the complexation of zinc ion with the ¹⁵N-labeled sensor, which allows the visualization of zinc ions via MRI. To create highly sensitive sensors with favorable relaxation properties, hydrogen atoms in proximity to the ¹⁵N nucleus were fully deuterated to mitigate the rapid relaxation induced by dipole-dipole interactions. This strategy effectively increases the lifetime of the hyperpolarization, which promises sufficient time for the subsequent zinc sensing. One merit of the current detection system stems from the slow ion complexation/release kinetics, which allows the quantification of zinc ions based on the integrations of ¹⁵N signals from free and zinc-complexed TPA. This feature is important to mitigate potential interferences and may lead to the simultaneous detection of multiple analytes with chemical-shift selective imaging. The manuscript is very well-written and the synthetic part is comprehensively described. The authors also demonstrated the utility of the current approach in the detection of zinc ions in BPH tissues. Therefore, I recommend its acceptance after addressing the following points.

1. What is the deuterium content in the TPA sensor? The proton residues are usually inevitable, how does it influence the lifetime of hyperpolarization?
2. The strategy to use deuterium to increase T₁ is documented and was implemented in ¹³C-based sensors. Relevant references should be included.
3. A linear relationship between measured and actual zinc ion concentrations is found in the range between 50 micro and 200 microM. Is quantification feasible for concentrations below 50 microM, where the integrations of ¹⁵N signals from free and zinc-complexed TPA differ significantly?
4. To detect human prostate tissue sample, a series of sample pretreatments, such as homogenization and centrifugation, were performed. These operations should be clarified in the main text to avoid confusions.

Reviewer #2 (Remarks to the Author):

In this report, Zoltan Kovacs and co-authors reported a new molecular probe for detection of Zn²⁺ using DNP-coupled ¹⁵N MRI. ¹⁵N-enriched and deuterated tris(2-pyridylmethyl)amine (¹⁵N,D-TPA) was designed and synthesized. ¹⁵N,D-TPA worked to detect Zn²⁺ ion in buffer and tissue homogenate. Advantages of this probe can be summarized as follows.

1. ¹⁵N,D-TPA was well hyperpolarized based on dissolution DNP process (6~8%P) and provided a highly enhanced NMR signals.
2. ¹⁵N-TPA binds Zn²⁺ to show 20 ppm change in ¹⁵N chemical shift, allowing easy discrimination of bound/unbound ¹⁵N signals even in the hyperpolarized state.
3. ¹⁵N,D-TPA, where dipole-dipole relaxations are minimized by reducing nearby ¹H, has relatively long T₁ (70.9 sec, and 57 sec with Zn).
4. HP ¹⁵N,D-TPA works as an agent for ratiometric ¹⁵N CSI, because it provides discriminable bound/unbound ¹⁵N signals.
5. HP ¹⁵N,D-TPA detects as low as 5 μM of Zn²⁺.
6. HP ¹⁵N,D-TPA was demonstrated to work as Zn²⁺ sensor in tissue homogenate.

The experiments were carefully designed and well carried out. The data largely support their conclusion that they have successfully developed a ¹⁵N DNP NMR/MRI sensor for the detection of Zn²⁺. However, some parts of this paper exaggerate and over-interpret the results. Overall, this includes sufficient advancement in DNP-MRI probe design and the paper deserves to be published in Communications Chemistry, but it needs to be revised according to the following comments.

Comments:

1. Detection of intracellular Zn²⁺:

The authors state that the advantage of this probe is its high cell permeability, thus allowing detection of freely available tissue Zn²⁺ in all compartments and not just Zn²⁺ ions released from cells. However, there is no data to validate this statement. In fact, they achieved detection of Zn²⁺ in tissue homogenates, but this does not guarantee that this probe detected cellular Zn²⁺ as well, since during the preparation of tissue homogenates, the cellular components should be destroyed.

The authors should demonstrate it or weaken this statement, just to simply say "detected Zn²⁺ in real tissue homogenate". In addition, "freely available Zn²⁺ in tissues" in title and "in human benign prostatic hyperplasia (BPH) tissue" in abstract are misleading and need to be changed.

2. How much is K_d of TPA-Zn²⁺ interaction? Depending on the value of K_d, the dynamic range of the sensor may change. It is necessary to discuss this point. It would also be useful for the reader to show how to calculate the concentration of Zn²⁺ in vivo where the concentration of the probe is not homogeneous (or to discuss the limitations in this regard).

3. Describe overall yields of 4 and 9 in main text.

4. Scheme 1: From compound 1 to 7, reagent (compound 6) is missing.

Reviewer #3 (Remarks to the Author):

In their manuscript, Suh and coworkers introduce the application of a ligand (TPA), hyperpolarized by means of d-DNP, for the detection and quantitation of freely available Zn²⁺ in tissues. The in-vivo detection of zinc (and other cations) using hyperpolarized ligands has already been proposed (see ref 23-25) but the advantage offered by the use of this molecule, with respect to those previously reported (¹³C-EGTA or ¹³C-cysteine), is its high selectivity and the fact that it allows direct quantification of freely available metal, that is not dependent on the concentration of the hyperpolarized probe.

In this proof-of-principle work, ¹⁵N-NMR spectra of the hyperpolarized ligand with different concentrations of Zn²⁺ were acquired, showing an excellent correlation between the concentration of the metal ion and the intensity of the ¹⁵N signal of the complex, with respect to the signal of the free ligand. ¹⁵N-chemical shift images have also been acquired on phantom solutions containing the free ligand and different concentrations of the free ion.

The authors synthesized the ¹⁵N-label tris(2-pyridylmethyl)amine ligand, in order to allow the detection of the ¹⁵N-hyperpolarized signals. The protons in the proximity of the ¹⁵N site were substituted with deuterium and the relaxation rate of the hyperpolarized signal decreased significantly, with respect to the fully protonated complex.

The feasibility of the detection of free Zn²⁺ in tissues was also tested using a homogenized prostate tissue sample. Concerning this experiment, we notice that the concentration of freely available Zn²⁺ in the homogenized tissue sample has been determined only by the ¹⁵N-spectra using d-DNP hyperpolarized ligand, but these might be validated using another method. This is important to strengthen the validity of the method.

All these reported tests demonstrate that, in principle, this d-DNP polarized ligand can allow the detection of free Zn²⁺. As far as the applicability of this hyperpolarized for the in-vivo detection of freely available Zn²⁺, some perplexities can be raised:

- The toxicity of the compound: the ligand concentration, for in-cells detection of zinc, is usually in the order of the tens of μM (20-80 μM), while MRS-MRI experiments require higher ligand concentration. In vitro experiments have been carried out using 1mM ligand (in-vitro samples) and 3.6mM on homogenized tissues, therefore the cytotoxicity of the compound, probably due to metal ions depletion, must be discussed, especially in the perspective of in-vivo applications.

- TPA is a membrane permeable Zn chelator, nevertheless, the rate of crossing of the cells membrane must be considered. The study carried out on a homogenized tissue sample show that the formation of the Zn²⁺ adduct is instantaneous in a complex matrix, but tissue homogenization might damage the cells membrane therefore the amount of detected zinc might not be intracellular. Experiments on intact cells would be useful, in order to add some information about

the rate of crossing the cells membrane by the TPA ligand.

In conclusion, this work is convincing and soundness is good. The relevance of this work in the community and in other fields depends much on the applicability of this method in-vivo, therefore we recommend a thorough revision of the manuscript in order to take into account all the raised criticisms.

Response to Reviewers (manuscript COMMSCHEM-20-0158-T)

Reviewer #1 (Remarks to the Author):

We thank the reviewer for the thorough review of our work and the valuable comments and the positive feedback. Please find below our point-by-point response to the Reviewer's concerns:

This manuscript by Kovacs and coworkers reports the detection of free available Zn²⁺ in tissues with a hyperpolarized ¹⁵N-labeled MRI sensor. A new and characteristic ¹⁵N NMR signal at precise chemical shift was produced upon the complexation of zinc ion with the ¹⁵N-labeled sensor, which allows the visualization of zinc ions via MRI. To create highly sensitive sensors with favorable relaxation properties, hydrogen atoms in proximity to the ¹⁵N nucleus were fully deuterated to mitigate the rapid relaxation induced by dipole-dipole interactions. This strategy effectively increases the lifetime of the hyperpolarization, which promises sufficient time for the subsequent zinc sensing. One merit of the current detection system stems from the slow ion complexation/release kinetics, which allows the quantification of zinc ions based on the integrations of ¹⁵N signals from free and zinc-complexed TPA. This feature is important to mitigate potential interferences and may lead to the simultaneous detection of multiple analytes with chemical-shift selective imaging. The manuscript is very well-written and the synthetic part is comprehensively described. The authors also demonstrated the utility of the current approach in the detection of zinc ions in BPH tissues. Therefore, I recommend its acceptance after addressing the following points.

1. What is the deuterium content in the TPA sensor? The proton residues are usually inevitable, how does it influence the lifetime of hyperpolarization?

The amount of residual ¹H can be estimated from the ¹H NMR spectra. Since we do not see any residual ¹H at ~3.5ppm (¹⁵NH₂CH₂), we are confident that the deuterium content in the TPA sensor is > 98% at least. This is expected as we started from commercially available deuterated reagents (i. e. we did not deuterate the final product).

2. The strategy to use deuterium to increase T1 is documented and was implemented in ¹³C-based sensors. Relevant references should be included.

We added the relevant references as requested (Ref 31-36).

3. A linear relationship between measured and actual zinc ion concentrations is found in the range between 50 micro and 200 microM. Is quantification feasible for concentrations below 50 microM, where the integrations of ¹⁵N signals from free and zinc-complexed TPA differ significantly?

Yes, it is feasible, but in this range, a lower concentration of HP-TPA may be used. Please, see also the added discussion about the quantification of Zn in the Supporting Information (S1. Quantification of Zn²⁺ concentration).

4. To detect human prostate tissue sample, a series of sample pretreatments, such as homogenization and centrifugation, were performed. These operations should be clarified in the main text to avoid confusions.

We thank the Reviewer for pointing this out. We clarified that the human tissue samples were homogenized for the HP- experiments (page 7).

Reviewer #2 (Remarks to the Author):

In this report, Zoltan Kovacs and co-authors reported a new molecular probe for detection of Zn²⁺ using DNP-coupled ¹⁵N MRI. ¹⁵N-enriched and deuterated tris(2-pyridylmethyl)amine (15N,D-TPA) was designed and synthesized. ¹⁵N,D-TPA worked to detect Zn²⁺ ion in buffer and tissue homogenate. Advantages of this probe can be summarized as follows.

- 1. ¹⁵N,D-TPA was well hyperpolarized based on dissolution DNP process (6~8%P) and provided a highly enhanced NMR signals.*
 - 2. ¹⁵N-TPA binds Zn²⁺ to show 20 ppm change in ¹⁵N chemical shift, allowing easy discrimination of bound/unbound ¹⁵N signals even in the hyperpolarized state.*
 - 3. ¹⁵N,D-TPA, where dipole-dipole relaxations are minimized by reducing nearby ¹H, has relatively long T₁ (70.9 sec, and 57 sec with Zn).*
 - 4. HP ¹⁵N,D-TPA works as an agent for ratiometric ¹⁵N CSI, because it provides discriminable bound/unbound ¹⁵N signals.*
 - 5. HP ¹⁵N,D-TPA detects as low as 5 μM of Zn²⁺.*
 - 6. HP ¹⁵N,D-TPA was demonstrated to work as Zn²⁺ sensor in tissue homogenate.*
- The experiments were carefully designed and well carried out. The data largely support their conclusion that they have successfully developed a ¹⁵N DNP NMR/MRI sensor for the detection of Zn²⁺. However, some parts of this paper exaggerate and over-interpret the results.*
- Overall, this includes sufficient advancement in DNP-MRI probe design and the paper deserves to be published in Communications Chemistry, but it needs to be revised according to the following comments.*
- Comments:*

We thank the reviewer for the thorough review of our work and the valuable comments and the positive feedback. Please find below our point-by-point response to the Reviewer's concerns:

1. Detection of intracellular Zn²⁺:

The authors state that the advantage of this probe is its high cell permeability, thus allowing detection of freely available tissue Zn²⁺ in all compartments and not just Zn²⁺ ions released from cells. However, there is no data to validate this statement. In fact, they achieved detection of Zn²⁺ in tissue homogenates, but this does not guarantee that this probe detected cellular Zn²⁺ as well, since during the preparation of tissue homogenates, the cellular components should be destroyed.

The authors should demonstrate it or weaken this statement, just to simply say “detected Zn²⁺ in real tissue homogenate”. In addition, “freely available Zn²⁺; in tissues” in title and “ in human benign prostatic hyperplasia (BPH) tissue” in abstract are misleading and need to be changed.

We thank the reviewer for raising this question. We agree with the reviewer that HP-studies with homogenized prostate tissue do not demonstrate the cell permeability of TPA. Therefore, we tested HP-TPA in intact human prostate epithelial (PNT1A) cells and added the result on page 7-8 (Figure 4). In these experiments, the extracellular Zn was washed away before the addition of the HP-sensor (see full description of the experiments in the Methods section).

2. How much is K_d of TPA-Zn²⁺; interaction? Depending on the value of K_d, the dynamic range of the sensor may change. It is necessary to discuss this point. It would also be useful for the reader to show how to calculate the concentration of Zn²⁺ in vivo where the concentration of the probe is not homogeneous (or to discuss the limitations in this regard).

We agree with the Reviewer that dissociation constant K_d of TPA-Zn²⁺ is an important parameter. Therefore, we included a discussion in the Supporting Information about the thermodynamic stability, dissociation constant and the dynamic range of the sensor as well as how the Zn-concentration was calculated (Quantification of Zn²⁺ concentration, S2).

3. Describe overall yields of 4 and 9 in main text.

The yields were added on page 4.

4. Scheme 1: From compound 1 to 7, reagent (compound 6) is missing.

We thank the reviewer for pointing this out. We fixed the scheme (Scheme 1).

Reviewer #3 (Remarks to the Author):

In their manuscript, Suh and coworkers introduce the application of a ligand (TPA), hyperpolarized by means of d-DNP, for the detection and quantitation of freely available Zn²⁺ in tissues. The in-vivo detection of zinc (and other cations) using hyperpolarized ligands has already been proposed (see ref 23-25) but the advantage offered by the use of this molecule, with respect to those previously reported (13C-EGTA or 13C- cysteine), is its high selectivity and the fact that it allows direct quantification of freely available metal, that is not dependent on the concentration of the hyperpolarized probe. In this proof-of-principle work, 15N-NMR spectra of the hyperpolarized ligand with different concentrations of Zn²⁺ were acquired, showing an excellent correlation between the concentration of the

metal ion and the intensity of the ^{15}N signal of the complex, with respect to the signal of the free ligand. ^{15}N -chemical shift images have also been acquired on phantom solutions containing the free ligand and different concentrations of the free ion.

The authors synthesized the ^{15}N -label tris(2-pyridylmethyl)amine ligand, in order to allow the detection of the ^{15}N -hyperpolarized signals. The protons in the proximity of the ^{15}N site were substituted with deuterium and the relaxation rate of the hyperpolarized signal decreased significantly, with respect to the fully protonated complex.

The feasibility of the detection of free Zn^{2+} in tissues was also tested using a homogenized prostate tissue sample. Concerning this experiment, we notice that the concentration of freely available Zn^{2+} in the homogenized tissue sample has been determined only by the ^{15}N -spectra using d-DNP hyperpolarized ligand, but these might be validated using another method. This is important to strengthen the validity of the method.

All these reported tests demonstrate that, in principle, this d-DNP polarized ligand can allow the detection of free Zn^{2+} . As far as the applicability of this hyperpolarized for the in-vivo detection of freely available Zn^{2+} , some perplexities can be raised:

- The toxicity of the compound: the ligand concentration, for in-cells detection of zinc, is usually in the order of the tens of μM (20-80 μM), while MRS-MRI experiments require higher ligand concentration. In vitro experiments have been carried out using 1mM ligand (in-vitro samples) and 3.6mM on homogenized tissues, therefore the cytotoxicity of the compound, probably due to metal ions depletion, must be discussed, especially in the perspective of in-vivo applications.

We thank the reviewer for the thorough review of our work and the valuable comments and the positive feedback. Please find below our point-by-point response to the Reviewer's concerns:

The toxicity of the compound:

We agree with the Reviewer that the toxicity of TPA may be a limitation. To the best of our knowledge, the LD_{50} value of TPA has not been reported. Cellular assays using HeLa cells showed that TPA was cytotoxic ($\text{LC}_{50} > 3\text{mM}$ in 1hr treatment and $\sim 38\mu\text{M}$ in 24hr treatment, Ref. 27) due to the interception of biological mobile zinc. But it should be noted that this is not necessary an indication of in vivo toxicity as intact animals have a much higher Zn-buffering capacity. Nevertheless, we included the following sentence on page 10 to highlight the importance of potential toxicity:

“The cytotoxicity of TPA due to its strong binding to Zn^{2+} , which can lead to zinc depletion, could limit its application for in vivo studies.²⁷ This, however, could be ameliorated by the design of second generation agents with weaker Zn-binding constant.”

- TPA is a membrane permeable Zn chelator, nevertheless, the rate of crossing of the cells membrane must be considered. The study carried out on a homogenized tissue sample show that the formation of the Zn^{2+} adduct is instantaneous in a complex matrix, but tissue homogenization might damage the cells membrane therefore the amount of detected zinc might not be intracellular. Experiments on intact cells

would be useful, in order to add some information about the rate of crossing the cells membrane by the TPA ligand.

We thank the Reviewer for raising this question. Tissue homogenization completely destroys cell membranes and thus, HP-studies with homogenized prostate tissue do not demonstrate the cell permeability of TPA. Therefore, we tested HP-TPA in intact cells and added the result on page 7-8 (Figure 4). In these experiments, the extracellular Zn was washed away before the addition of the HP-sensor (see full description of the experiments in the Methods section).

In conclusion, this work is convincing and soundness is good. The relevance of this work in the community and in other fields depends much on the applicability of this method in-vivo, therefore we recommend a thorough revision of the manuscript in order to take into account all the raised criticisms.

REVIEWERS' COMMENTS:

Reviewer #2 (Remarks to the Author):

I am satisfied with this revision.

Reviewer #3 (Remarks to the Author):

The manuscript by Dr. Kovacs and collaborators has been carefully revised and all the raised criticisms have been addressed.

This is a very nice work and it certainly worth publication in Commun. Chemistry